# Metabolic Stability of New Mito-Protective Short-Chain Naphthoquinones

**DOI:** 10.3390/ph13020029

**Published:** 2020-02-12

**Authors:** Zikai Feng, Jason A. Smith, Nuri Gueven, Joselito P. Quirino

**Affiliations:** 1Australian Centre for Research on Separation Sciences (ACROSS), School of Natural Sciences-Chemistry, College of Science and Engineering, University of Tasmania, Hobart, TAS 7005, Australia; zikai.feng@utas.edu.au; 2School of Natural Sciences-Chemistry, College of Science and Engineering, University of Tasmania, Hobart, TAS 7005, Australia; jason.smith@utas.edu.au; 3Division of Pharmacy, School of Medicine, College of Health and Medicine, University of Tasmania, Hobart, TAS 7005, Australia; nuri.guven@utas.edu.au

**Keywords:** mitochondrial dysfunction, idebenone, short-chain quinone, metabolic stability, HepG2 cell culture, reverse-phase liquid chromatography

## Abstract

Short-chain quinones (SCQs) have been identified as potential drug candidates against mitochondrial dysfunction, which is largely dependent on their reversible redox characteristics of the active quinone core. We recently synthesized a SCQ library of > 148 naphthoquinone derivatives and identified 16 compounds with enhanced cytoprotection compared to the clinically used benzoquinone idebenone. One of the major drawbacks of idebenone is its high metabolic conversion in the liver, which significantly restricts its therapeutic activity. Therefore, this study assessed the metabolic stability of the 16 identified naphthoquinone derivatives **1**–**16** using hepatocarcinoma cells in combination with an optimized reverse-phase liquid chromatography (RP-LC) method. Most of the derivatives showed significantly better stability than idebenone over 6 hours (*p* < 0.001). By extending the side-chain of SCQs, increased stability for some compounds was observed. Metabolic conversion from the derivative **3** to **5** and reduced idebenone metabolism in the presence of **5** were also observed. These results highlight the therapeutic potential of naphthoquinone-based SCQs and provide essential insights for future drug design, prodrug therapy and polytherapy, respectively.

## 1. Introduction

Mitochondrial dysfunction has been linked to a vast number of disorders ranging from primary mitochondrial disorders such as Leber’s Hereditary Optical Neuropathy (LHON) to common diseases associated with mitochondrial dysfunction such as diabetes [1,2,3]. Despite the large numbers of patients that show mitochondrial dysfunction, there are hardly any drugs on the market that aim to target mitochondrial function directly. This represents a significant unmet medical need and thus, new drug candidates are needed that can be developed for this purpose. Potential drugs to protect against mitochondrial dysfunction include short chain quinones (SCQs), which possess reversible redox characteristics due to the quinone core [4,5,6]. The benzoquinone idebenone has shown some limited activity to protect against vision loss and restore visual acuity in patients with LHON [7,8,9]. As a consequence, it has been marketed in Europe for this purpose since 2015. In line with a protective activity against mitochondrial dysfunction, it was recently suggested that idebenone could also have anti-diabetic activity based on an insulin-sensitizing effect. This activity was suggested to be based on its ability to inhibit the interaction of p52Shc with the insulin receptor [10].

We recently reported the design and synthesis of a library of > 148 short-chain naphthoquinone derivatives [11] that intended to overcome the known limitations of idebenone such as limited bioactivation and rapid metabolic inactivation. From this panel, 16 SCQs (**1**-**16**, Table 1) showed significantly improved cytoprotective activity in vitro compared to idebenone (*p* < 0.033) under conditions of mitochondrial dysfunction [11]. The current study determined the in vitro metabolic stability of 16 new SCQs to identify promising drug development candidates, before in vivo pharmacokinetic studies in animal models can be initiated. A simple and efficient analytical methodology was developed based on gradient-elution reverse-phase liquid chromatography (RP-LC) in conjunction with sample preparation by acetonitrile (ACN) precipitation. This methodology allowed for the required quantitation of SCQs at appropriate μM concentrations in the highly complex cell culture media used in in vitro metabolic stability studies. The human hepatic cell line HepG2 was previously described to mimic in vivo metabolism with liver-like conditions [12]. Although HepG2 cells show lower expression of metabolic enzymes compared to human liver samples ex vivo, this cell line is perfectly suited and widely used for in vitro metabolic studies due to their high phenotypic stability and unlimited availability, which provides a robust and reproducible test platform [13,14]. This study was essential to anticipate drug behavior (pharmacokinetics and metabolism) in vivo and will aidin the development of the most promising compounds toward their clinical use.

## 2. Materials and Methods

### 2.1. Chemicals, Solutions, and Cells

Idebenone was provided by Santhera Pharmaceuticals (Pratteln, Switzerland) as a reference compound. The novel SCQs (**1**–**16**) were synthesized as described previously [11]. LC-grade ACN was purchased from VWR (Queensland, Australia). Purified water was from a Milli-Q system (Millipore, MA, USA). Formic acid (FA), dimethylsulfoxide (DMSO), Dulbecco Modified Eagle Medium (DMEM, D5523), and sodium bicarbonate were purchased from Sigma-Aldrich (New South Wales, Australia). Fetal bovine serum (FBS) was purchased from SAFC Biosciences (Victoria, Australia). 0.25% Trypsin, ethylenediaminetetraacetic acid (EDTA), and phosphate-buffered saline (PBS) tablets were purchased from ThermoFisher Scientific (Victoria, Australia). The HepG2 cell line (HB-8065) was purchased from ATCC (Manassas, VA, USA).

The stock solutions of each SCQ (100 mM) were prepared in DMSO and stored at −20 °C. Working standard solutions of each SCQ (1.0 μM, 2.5 μM, 5.0 μM, 7.5 μM, 10.0 μM) were prepared by dilution of the appropriate stock solutions with 25% ACN in water. DMEM cell culture media was prepared according to the manufactures instructions and sterilized by filtration using 0.22 μm bottle top filters (Corning, VIC, AU). DMEM was supplemented with FBS (10%), sodium bicarbonate (3.7 g L^−1^), and stored at 4 °C. EDTA solution (0.5 mM, pH 8) was sterilized using 0.45 μm filters and stored at −20 °C.

### 2.2. RP-LC Instrumentation

The LogP values in Table 1 were predicted by the ChemDraw software (PerkinElmer, Waltham, MA, USA). The LogP values suggested that these SCQs range from low (0.74) to intermediate (3.50) lipophilicity. Thus, RP-LC was used for separation and quantitation. An UltiMate™ 3000 LC system equipped with an UV detector (ThermoFisher Scientific, Victoria, Australia) was used. The system operation, data acquisition and processing were performed using Chromeleon software (version 6.0, ThermoFisher Scientific, Victoria, Australia). Analytical separations were carried out on an Acclaim™ Polar Advantage II RP-LC column (2.2 μm, 2.1 × 10 mm) at 25 °C. Mobile phase A and B was 0.1% FA in purified water and 0.1% FA in ACN, respectively. Flow rate was 0.2 mL min^−1^ which gave a void time (VT) of 1.2 min.

### 2.3. Metabolic Stability Study

#### 2.3.1. Cell Culture

HepG2 cells were cultured in DMEM culture media in an atmosphere of 95% humidified air and 5% CO_2_ at 37 °C. Cells were passaged twice a week when reaching approximately 75% confluency. Cell monolayers were washed with PBS once before harvested with 0.5 mL EDTA and 0.5 mL trypsin for 3.5 min and counted using a hemocytometer (Paul Marienfeld GmbH, Lauda-Königshofen, Germany). Cells were routinely grown in 25 cm^2^ cell culture flasks (2 μm vent cap, Corning, Victoria, Australia) with 2 × 10^6^ cells seeded in 5 mL culture media. After thawing from liquid nitrogen storage, cells were passaged for at least 2–3 weeks to reach steady cumulative growth rates before used for any experiments.

#### 2.3.2. Cell Culture System Development

To effectively compare the metabolic stability of our novel SCQs to the reference compound idebenone, a cell culture system had to be developed that would replicate the high metabolic conversion of idebenone in the liver observed in vivo [15,16,17]. We therefore employed the hepatic cell line HepG2 to expose the test compounds to hepatic-like enzymatic activities [8]. In addition, the test concentration for SCQs and idebenone had to take into consideration two opposing factors. On the one hand the assay would need to contain sufficient test compound to maximize detection accuracy, on the other hand, the concentration needed to be below the levels that might induce toxicity towards the cultured cells. Based on previous data [11], no significant cell loss was observed when HepG2 cells were treated with most SCQs at a concentration of 200 μM for 24 h, which led us to select the lower concentration of 40 μM as test concentration. In addition, to approach the metabolic conversion rates of idebenone in vivo (***t***_1/2_ = 3 h at a single dose of 150 mg) [15], three different cell densities of 2.5 × 10^5^, 5.0 × 10^5^ and 1.0 × 10^6^ cells in 2 mL media were initially evaluated using a single concentration of idebenone (40 μM). All three cell densities showed similar rates of metabolic conversion of idebenone with about ~50%, 76% and 87% metabolized drug after 2 h, 4 h and 6 h, respectively (Appendix A). We therefore selected the lowest cell concentration (2.5 × 10^5^ cells in 2 mL culture media) for this assay.

#### 2.3.3. Sample Preparation

Log phase HepG2 cells were seeded at 2.5 × 10^5^ cells well^−1^ in tissue culture-treated 6-well plates (Corning, Victoria, Australia) and allowed to adhere overnight. After one day, the culture media was replaced with fresh culture media (2 mL well^−1^) containing the test compounds (12 parallel wells per SCQ). After incubation of cells with test compounds for up to 6 hours, the cell culture media containing the residual SCQs was removed from the six-well plates. Then, 1 mL of cell culture media containing each tested compound was collected at different time points (***t*** = 0, 2 h, 4 h, 6 h). Several methods of sample preparation were tested (e.g., dilution with organic solvents followed by evaporation) and the most suited approach was identified. Each sample (1 mL) was mixed 1:1 with ACN, vortexed for 10 s, and then centrifuged at 2000× *g* for 10 min at 25 °C. 1 mL of each supernatant was diluted 1:1 in purified water and filtered using 0.45 μm filters prior to immediate analysis by RP-LC. The final concentration of tested compounds for RP-LC analysis was ≤10 μM.

#### 2.3.4. RP-LC Gradient Optimization and Analytical Performance

The RP-LC method was developed so that ≤10 μM of each SCQ could be quantified with acceptable repeatability (RSD%) and recovery (%). The mobile phase gradient was adjusted to generate a retention time (RT) of all compounds between 3.00–9.05 min (Appendix A). The final conditions included a gradient flow of mobile phase B: 25% (same ACN% in the sample) for 2 min, 25–95% for 3 min, 95% for 4 min, 95–25% for 1 min, and 25% for 5 min (total run ***t*** = 15 min, including column post-conditioning). The injection volume (2–20 μL) and detection wavelength (210 nm, 230 nm, 254 nm, 480 nm) were then varied to optimize sensitivity, with 20 μL sample injection volume and detection at 210 nm identified as ideal (Appendix A). Appendix A summarizes the analytical figures of merit for the RP-LC of all the compounds. Peak areas were found to be linear between 1–10 μM (Appendix A), with coefficients of determination (R2) between 0.982–1.000. The limit of quantification (LOQ) of each compound was established as 1 μM. RSD% (n = 3) at all concentrations (1.0 μM, 2.5 μM, 5.0 μM, 7.5 μM, and 10 μM) were between 0.4–7.6%. The percentage recovery (86.7–116.2%) was calculated by dividing the concentrations found at ***t*** = 0 by 10 μM × 100%, with RSD% from 1.2–10.6%. The above values for analytical figures of merit were considered acceptable for determining metabolic stability of the compounds with concentrations ≤10 μM.

### 2.4. Statistical Analysis

GraphPad Prism (version 8.2.1, San Diego, CA, USA) was used to perform statistical analysis between three or more groups through one-way or two-way ANOVA, respectively. The differences were statistically significant when *** *p* < 0.001, ** *p* < 0.002, * *p* < 0.033.

## 3. Results and Discussion

### 3.1. Superior Metabolic Stability of UTAS SCQs 

The recovery (%) of SCQs at ***t*** = 0 was very consistent by using precipitation with ACN (Figure 1). Idebenone showed a significant reduction from ***t*** = 2 h onwards (*p* < 0.001) with ~27.3% remaining at ***t*** = 6 h, which was consistent with its short half-life in vivo [15,18]. In contrast, of the 16 SCQs tested, six (**1**, **2**, **4**, **5**, **6** and **10**) demonstrated supreme stability without a significant metabolic conversion over a period of 6 hours. Except for the sulfide derivative **16**, most of the SCQs tested were significantly more stable than idebenone (*p* < 0.001). At ***t*** = 4 h and 6 h, 15 SCQs (excluding **16**) were significantly more stable than idebenone (**7**
*p* < 0.033; others *p* < 0.001). Given that the enzymatic activities of HepG2 cells are not comparable to those of fresh liver samples or other immortal cell lines such as HepaRG [13,14], it has to be noted that HepG2 cells may not be sensitive enough to differentiate our best six SCQs **1**, **2**, **4**, **5**, **6** and **10**. On the other hand, the comparative metabolic kinetics against idebenone using our test system clearly demonstrated improved stability of the novel SCQs. Although the use of fresh liver biopsies or different cell lines could have increased the speed of metabolic conversion in vitro, it is important to point out that the kinetics achieved in our test system mirror the metabolism of idebenone in patients with reported ***t***_1/2_ of ~3 h in vivo (150 mg in a single dose) [15]. In addition, our results highlight that the sulfide derivative **16** is not as competitive as the other amides **1**–**15**, due to its significantly reduced metabolic stability.

### 3.2. Increased Metabolic Stability by Carbon Chain Extension

Apart from the metabolic stability information provided by this assay, structure-metabolic stability relationships were also obtained from the 16 compounds. The results indicate that the amide linkage is very stable in general and instability occurs due to other substituents in the side chain. Comparing the difference in amides, the data suggest that the carbon chain between the quinone core and the amide linkage appears to increase metabolic stability (Figure 2). The tyramine derivative **7** was less stable, likely due to the phenolic group, but the tyramine derivatives **8** and **9** showed increased metabolic stability when the carbon chain was extended. **9** was significantly more stable than **8** over 4 h (*p* < 0.002), 6 h (*p* < 0.033) and **7** at all time points (*p* < 0.033). This increased stability correlates with increased lipophilicity, as it correlates to the log of distribution coefficient D (LogD) with increases from 3.43, 3.87 to 4.31. Increased lipophilicity could either increase affinity to its target and to cellular and mitochondrial membranes or at higher levels could reduce absorption due to a higher membrane localization. In the current test system this would reduce interaction with metabolic enzymes located in the cytoplasm but in vivo could reduce the absorption of the compounds or their blood-brain barrier penetration. This connection between lipophilicity and metabolic stability was not observed for the *L*-phenylalanine derivatives **5** and **6**, which were two of the six most stable compounds.

### 3.3. Natural Enantiomer as a Prodrug Alternative

While no differences between the stable enantiomers **1** and **2** were observed, the naturally occurring *L*-phenylaninol **3** was found much less stable than its unnatural *D*-enantiomer **4** (*p* < 0.001) (Figure 3). This indicates some selectivity for the enzymatic degradation of this enantiomer. In addition, the *L*-phenylalaninol derivative **3** was much less stable than and its oxidized form, the *L*-phenylalanine derivative **5** (*p* < 0.001). From the *L*-prolinol derivative **14** to its oxidized form, the *L*-proline derivative **15**, the change in stability did not reach statistical significance. Initially, these two oxidized forms were expected to be the metabolites of the two reduced forms, respectively. The peaks of the oxidized forms were expected to be observed and quantifiable from chromatograms of the reduced forms according to relevant retention times and linearity. So far, a ~10% conversion from the *L*-phenylalaninol derivative **3** to the *L*-phenylalanine derivative **5** was detected according to the retention time, yet conversion from **14** towards **15** was not detected at all. Furthermore, the conversion from **3** to **5** was confirmed using mass spectrometry (Appendix A). In comparison with their enantiomers, no conversion was detected for the unnatural *D*-phenylalaninol derivative **4** to its oxidized form *D*-phenylalanine derivative **UTAS#94** (Figure 3) [11], which was not cytoprotective enough to be selected for the current study. This suggested that the reduced form **3** might be used as a prodrug for the oxidized form **5** as an alternative. Given the lower lipophilicity of **5** (LogD = 0.12), **3** (LogD = 3.10) could represent a promising prodrug approach where upon crossing the blood-brain barrier, conversion from **3** to **5** would produce a highly active drug candidate to protect against mitochondrial dysfunction-induced neurotoxicity and visual impairments.

### 3.4. Metabolically Stable UTAS SCQ as an Alternative for Polytherapy

The conversion from the *L*-phenylalaninol derivative **3** to the *L*-phenylalanine derivative **5** appeared to reach a plateau at ~10% conversion from ***t*** = 2 h and persisted for the entire test period until ***t*** = 6 h. Guided by the superior stability of **5**, we hypothesized that **5** might be a metabolic enzyme inhibitor. A concentration series of the unstable reference SCQ idebenone was tested for metabolic stability for 6 h with or without 40 μM **5** (Figure 4). A significant reduction of idebenone metabolism by 40 μM **5** was observed from 20 μM idebenone onward. Not surprisingly, **5** was found to be stable, as described in combination with all concentrations of idebenone (Appendix A). These results suggested that **5** inhibited metabolic enzymes and could be used in combination with idebenone as a polytherapy alternative to overcome its poor stability reported in vivo [15,18]. Future studies will need to address the type of inhibition and to identify the specific enzyme that is inhibited.

## 4. Conclusions

A library of novel short-chain naphthoquinones was designed to support the discovery of drug candidates to protect against mitochondrial dysfunction [11]. From this library, 16 compounds showed significantly improved cytoprotective activity under conditions of mitochondrial dysfunction. We presented new methods to study the metabolic stability of our novel SCQs in vitro. Our method is characterized by high recovery rates, due to a simple precipitation procedure that eliminated interferences during RP-LC analysis. The methods also provided quick and reliable results in an accelerated manner within a typical working day. This allowed us to mimic the metabolic conversion of SCQs in vitro in a manner comparable to their metabolic conversion in vivo. Among 16 new SCQs tested, 15 showed significant metabolic stability compared to the clinically used benzoquinone idebenone. Furthermore, structure-metabolic stability relationships, metabolic conversions and inhibition of idebenone metabolism were addressed. Overall, these methods and results not only assist to anticipate drug behavior in vivo in terms of their pharmacokinetic properties and metabolism but also provide essential insights for future drug design, prodrug therapy and polytherapy.

## Figures and Tables

**Figure 1 pharmaceuticals-13-00029-f001:**
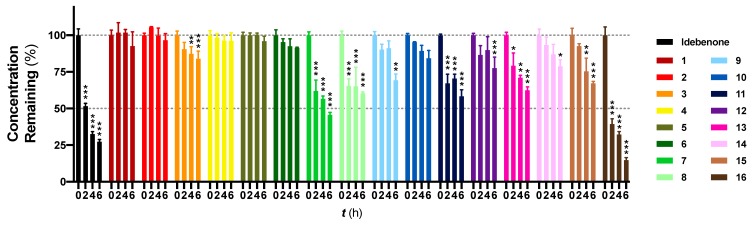
Metabolic stability study of 16 new SCQs and the reference benzoquinone idebenone. HepG2 cells were exposed to each SCQ at 40 μM for up to 6 h. Concentrations found at ***t*** = 0 were normalized to 100% and concentrations found at other time points were normalized by accordingly dividing by the concentrations found at ***t*** = 0, × 100%. Data was expressed as mean ± standard error of mean (SEM) from at least one experiment, with at least three data points each. Two-way ANOVA was performed to compare concentrations found at ***t*** = 2 h, 4 h, 6 h to ***t*** = 0 for each compound: *** *p* < 0.001, ** *p* < 0.002, * *p* < 0.033.

**Figure 2 pharmaceuticals-13-00029-f002:**
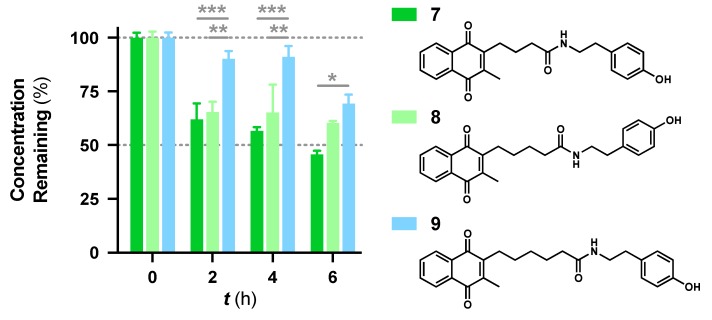
Comparison of metabolic stability and carbon chain length between the quinone core and the amide linkage for the structurally similar tyramine derivatives **7**–**9**. Data was expressed as mean ± SEM from at least one experiment, with at least three data points each: *** *p* < 0.001, ** *p* < 0.002, * *p* < 0.033.

**Figure 3 pharmaceuticals-13-00029-f003:**
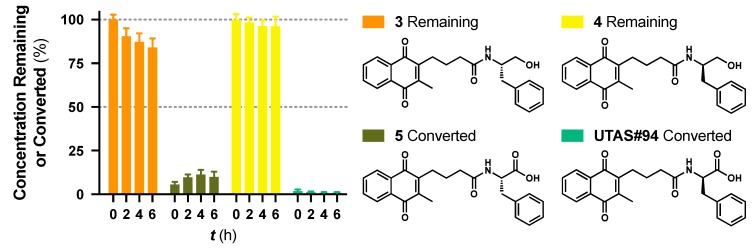
Metabolic stability of enantiomers **3** and **4** and their metabolic conversion to their oxidized forms **5**. and **UTAS#94**, respectively. The concentration of **5** and **UTAS#94** were calculated according to their analytical figures of merit, followed by normalization over the initial recovered concentrations of **3** and **4**, respectively. Data was expressed as mean ± SEM from three independent experiments, with at least 3 data points each.

**Figure 4 pharmaceuticals-13-00029-f004:**
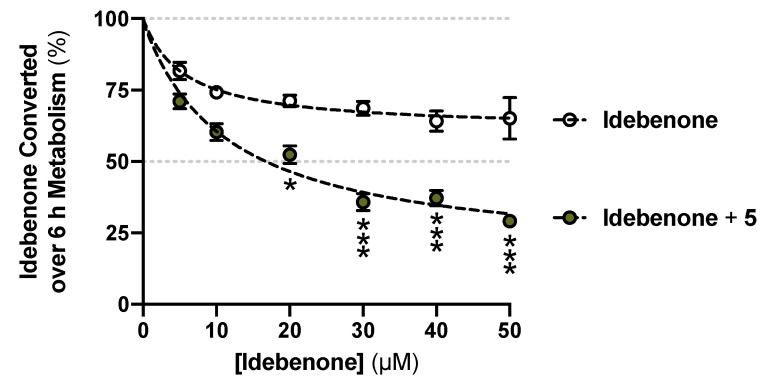
Reduced idebenone metabolism in the presence of the *L*-phenylalanine derivative **5**. Data was expressed as mean ± SEM from three independent experiments, with 4 data points each. Non-linear fits and two-way ANOVA were performed for comparisons between with or without **5** supplemented: *** *p* < 0.001, * *p* < 0.033.

**Table 1 pharmaceuticals-13-00029-t001:** Chemical structure, physical properties and in vitro efficacy of 16 novel short-chain quinones (SCQs) and the reference compound idebenone.

Compound	ID	Structure	n	R	Formula	Molecular Weight (g mol^−1^)	LogP ^1^	LogD ^2^	In vitro Cytoprotection (%) ^3^
**Idebenone**		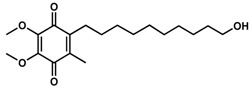			C_19_H_30_O_5_	338.4	1.24	3.57	66.2 ± 12.0
**1**	UTAS#81	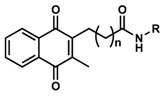	2	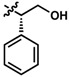	C_23_H_23_NO_4_	377.4	2.24	2.81	83.8 ± 19.9
**2**	UTAS#80	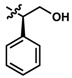	C_23_H_23_NO_4_	377.4	2.24	2.81	87.6 ± 19.7
**3**	UTAS#62	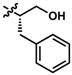	C_24_H_25_NO_4_	391.5	2.52	3.10	93.1 ± 13.7
**4**	UTAS#78	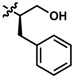	C_24_H_25_NO_4_	391.5	2.52	3.10	80.0 ± 21.1
**5**	UTAS#37	2	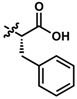	C_24_H_23_NO_5_	405.4	2.48	0.12	100.3 ± 17.3
**6**	UTAS#72	3	C_25_H_25_NO_5_	419.5	2.90	0.74	90.7 ± 15.6
**7**	UTAS#74	2	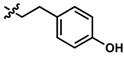	C_23_H_23_NO_4_	377.4	2.67	3.43	91.7 ± 15.6
**8**	UTAS#88	3	C_24_H_25_NO_4_	391.5	3.09	3.87	91.8 ± 9.8
**9**	UTAS#89	4	C_25_H_27_NO_4_	405.5	3.50	4.31	85.2 ± 9.5
**10**	UTAS#54	2	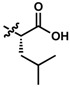	C_21_H_25_NO_5_	371.4	2.04	0.26	98.7 ± 10.9
**11**	UTAS#77	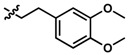	C_25_H_27_NO_5_	421.5	2.80	3.41	95.9 ± 19.4
**12**	UTAS#91	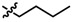	C_19_H_23_NO_3_	313.4	2.29	3.04	82.0 ± 7.1
**13**	UTAS#95	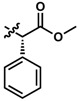	C_24_H_23_NO_5_	405.4	2.46	3.28	86.1 ± 5.0
**14**	UTAS#61	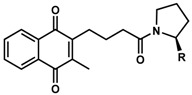		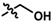	C_20_H_23_NO_4_	341.4	1.06	1.71	100.7 ± 28.4
**15**	UTAS#43		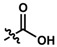	C_20_H_21_NO_5_	355.4	1.02	−1.32	92.7 ± 7.6
**16**	UTAS#46	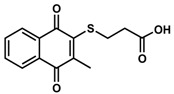			C_14_H_12_O_4_S	276.3	0.74	−1.43	80.5 ± 11.5

^1^ LogP was predicted using ChemDraw Professional software (version 16.0, PerkinElmer, Waltham, MA, USA). ^2^ LogD was predicted using MarvinView software (version 19.25, ChemAxon, Budapest, Hungary). ^3^ In vitro cytoprotection of HepG2 by 10 μM SCQs against rotenone-induced mitochondrial complex I dysfunction. Cytoprotection was calculated as a relative percentage of cell survival compared to untreated cells (27.9 ± 7.9%). Data was expressed as mean ± standard deviation (SD) [11].

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
