# Peer review of "Metabolic Stability of New Mito-Protective Short-Chain Naphthoquinones"

_pharmaceuticals, 2020, doi:10.3390/ph13020029_

Round 1

Reviewer 1 Report

Manuscript entitled, 'Metabolic stability of new mito-protective short chain naphthoquinones" by Zikai Feng et al. is well written article.  The authors have identified 16 short chained naphthoquinone compounds from a list of 148 analogs having enhanced cytoprotection compared to standard agent, Idebenone.  The newly identified agents are inhibitors of mitochondrial dysfunctions.  Among 16 identified agents,  15 compounds have shown better metabolic stability, structural-metabolic stability and developed quick and reliable method of detection by using Reverse Phase Liquid Chromatography . 

The analysis and results presented here are all from in vitro studies and conclusion need to confirm by in vivo before generalizing the newly developed method and stability data of naphthoquinone analogs.

In addition, there a mistake in line 81 need to be corrected as LogP values will not give solubility prediction.  The line 81 need to read as LogD values suggested.....     

Author Response

Dear reviewer,

We would like to express our gratitude to your comments and feedback. In the following, we respond to your comments and feedback point-by-point.

Point 1: “English language and style are fine/minor spell check required”.

Response 1:

Line 13: hyphen added between “Short” and “chain” Line 164: “p” font style changed to italics Line 221: hyphen deleted to make “prodrug” single word Line 244: round brackets changed to box brackets for referencing use

Point 2: “line 81 needs to be corrected as LogP values will not give solubility prediction”.

Response 2: Thank you very much for pointing out this error. In Line 93 (error mentioned is now in Line 93 after revision), “solubility” has been corrected to “lipophilicity”. The sentence now reads “The LogP values suggested that these SCQs range from low (0.74) to intermediate (3.50) lipophilicity”.

In addition, all four figures (Line 181, 189, 226, 247) in the manuscript have now been replaced in eps format for best resolution. Please do not hesitate to contact us if you have any additional questions.

We sincerely wish you a happy New Year.

Warm regards,

Zikai Feng, Jason A. Smith, Nuri Gueven & Joselito P. Quirino

Reviewer 2 Report

Although authors described and discussed their results clearly, this manuscript have major points that should be discussed.

Authors only showed the chemical stability of 16 compounds in HepG2 cells without showing in vitro efficacy of these compounds. Please add the efficacy results of these compounds to show their potential as drug candidates. Authors used HepG2 cells to investigate the metabolic stability of 16 compounds. However, HepG2 cells showed limited expression of metabolic enzymes compared with cryopreserved hepatocytes and HepaRG cells therefore, this system has the limitation to mimic the metabolic stability and hepatotoxicity. Authors should point out this limitation in the discussion.   

References.  

(1) Characterization of primary human hepatocytes, HepG2 cells, and HepaRG cells at the mRNA level and CYP activity in response to inducers and their predictivity for the detection of human hepatotoxins. Gerets HH1, Tilmant K, Gerin B, Chanteux H, Depelchin BO, Dhalluin S, Atienzar FA. Cell Biol Toxicol. 2012 Apr;28(2):69-87. doi: 10.1007/s10565-011-9208-4. Epub 2012 Jan 19.

(2) Comparison of Drug Metabolism and Its Related Hepatotoxic Effects in HepaRG, Cryopreserved Human Hepatocytes, and HepG2 Cell Cultures. Yokoyama Y1,2, Sasaki Y1, Terasaki N1, Kawataki T1, Takekawa K1, Iwase Y1, Shimizu T1, Sanoh S2, Ohta S2. Biol Pharm Bull. 2018 May 1;41(5):722-732. doi: 10.1248/bpb.b17-00913. Epub 2018 Feb 14.

Author Response

Dear reviewer,

We would like to express our gratitude to your comments and feedback. In the following, we respond to your comments and feedback point-by-point.

Point 1: “Authors only showed the chemical stability of 16 compounds in HepG2 cells without showing in vitro efficacy of these compounds. Please add the efficacy results of these compounds to show their potential as drug candidates.”

Response 1: Thank you very much for this comment. We previously assessed the in vitro efficacy of these compounds (Woolley et al., 2019). The efficacy results (in vitro cytoprotection of HepG2) have now been referenced and shown in Table 1.

Table 1 title (Line 47) has been changed to “Chemical structure, physical properties and in vitro efficacy of 16 novel SCQs and the reference compound idebenone.” Explanations for “LogP”, “LogD” and “In vitro Cytoprotection” have now been added as footnotes (Line 55-58): “1 LogP was predicted using ChemDraw Professional software (version 16.0, PerkinElmer, MA, USA). 2 LogD was predicted using MarvinView software (version 19.25, ChemAxon, Budapest, Hungary). 3 In vitrocytoprotection of HepG2 by 10 μM SCQs against rotenone-induced mitochondrial complex I dysfunction. Cytoprotection was calculated as a relative percentage of cell survival compared to untreated cells (33.6±11.1 %). Data was expressed as mean ± standard deviation (SD)”. The main text (Line 45, 59-60) has been reworded as “From this panel, 16 SCQs (1-16, Table 1) showed significantly improved in vitro cytoprotective activity compared to idebenone (p < 0.033) under conditions of mitochondrial dysfunction [11].”

Reference:

Woolley, K.L.; Nadikudi, M.; Koupaei, M.N.; Corban, M.; McCartney, P.; Bissember, A.C.; Lewis, T.W.; Gueven, N.; Smith, J.A. Amide linked redox-active naphthoquinones for the treatment of mitochondrial dysfunction. MedChemComm 2019, 10, 399-412, doi:10.1039/C8MD00582F.

Point 2: “Authors used HepG2 cells to investigate the metabolic stability of 16 compounds. However, HepG2 cells showed limited expression of metabolic enzymes compared with cryopreserved hepatocytes and HepaRG cells therefore, this system has the limitation to mimic the metabolic stability and hepatotoxicity. Authors should point out this limitation in the discussion.”

Response 2: Thank you very much for pointing out this limitation. The limited expression of metabolic enzymes in HepG2 has been addressed and referenced in Introduction, Results and Discussion 3.1:

Introduction (Line 68-70): “... Comparing to fresh human liver samples, though HepG2 was less suited for metabolic studies due to limited expression of metabolic enzymes, it was ideal for this study because of their high phenotypic stability, unlimited availability and ease to obtain [13-14].’’ Results and Discussion 3.1 (Line 169-172) has been reworded as: “... Given that the enzymatic activities in HepG2 are not as comparable as that in fresh liver samples or other immortal cell lines such as HepaRG [13-14], HepG2 may not be sensitive enough to differentiate our best six SCQs 1, 2, 4, 5, 6 and 10. Still, these results sufficiently indicate that the sulfide derivative 16 is not as competitive as the other amides 1-15 for the development of our SCQ, due to its poorer metabolic stability.’’

We would like to thank you again for this comment, as cryopreserved hepatocytes and HepaRG cell line will definitely aid the future in vitro metabolic studies of our best compounds towards their clinical use.

References:

Gerets, H.H.J.; Tilmant, K.; Gerin, B.; Chanteux, H.; Depelchin, B.O.; Dhalluin, S.; Atienzar, F.A. Characterization of primary human hepatocytes, HepG2 cells, and HepaRG cells at the mRNA level and CYP activity in response to inducers and their predictivity for the detection of human hepatotoxins. Cell Biol Toxicol 2012, 28, 69-87, doi:10.1007/s10565-011-9208-4.

Yokoyama, Y.; Sasaki, Y.; Terasaki, N.; Kawataki, T.; Takekawa, K.; Iwase, Y.; Shimizu, T.; Sanoh, S.; Ohta, S. Comparison of Drug Metabolism and Its Related Hepatotoxic Effects in HepaRG, Cryopreserved Human Hepatocytes, and HepG2 Cell Cultures. Biol Pharm Bull 2018, 41, 722-732.We

In addition, all four figures (Line 181, 189, 226, 247) in the manuscript have now been replaced in eps format for best resolution. Please do not hesitate to contact us if you have any additional questions.

We sincerely wish you a happy New Year.

Warm regards,

Zikai Feng, Jason A. Smith, Nuri Gueven & Joselito P. Quirino

Round 2

Reviewer 2 Report

Ms was revised according to the reviewer's comments